# Biofilms and Benign Colonic Diseases

**DOI:** 10.3390/ijms232214259

**Published:** 2022-11-17

**Authors:** Busara Songtanin, Christopher J. Peterson, Adebayo J. Molehin, Kenneth Nugent

**Affiliations:** 1Department of Internal Medicine, Texas Tech University Health Sciences Center, 3601 4th Street, Lubbock, TX 79430, USA; 2Department of Internal Medicine, Virginia Tech Carilion School of Medicine, Roanoke, VA 24016, USA; 3Department of Microbiology & Immunology, College of Graduate Studies, Midwestern University, Glendale, AZ 85308, USA

**Keywords:** biofilm, microbiome, interstitial epithelial cells, nutrition, innate immunity

## Abstract

The colon has a very large surface area that is covered by a dense mucus layer. The biomass in the colon includes 500–1000 bacterial species at concentrations of ~10^12^ colony-forming units per gram of feces. The intestinal epithelial cells and the commensal bacteria in the colon have a symbiotic relationship that results in nutritional support for the epithelial cells by the bacteria and maintenance of the optimal commensal bacterial population by colonic host defenses. Bacteria can form biofilms in the colon, but the exact frequency is uncertain because routine methods to undertake colonoscopy (i.e., bowel preparation) may dislodge these biofilms. Bacteria in biofilms represent a complex community that includes living and dead bacteria and an extracellular matrix composed of polysaccharides, proteins, DNA, and exogenous debris in the colon. The formation of biofilms occurs in benign colonic diseases, such as inflammatory bowel disease and irritable bowel syndrome. The development of a biofilm might serve as a marker for ongoing colonic inflammation. Alternatively, the development of biofilms could contribute to the pathogenesis of these disorders by providing sanctuaries for pathogenic bacteria and reducing the commensal bacterial population. Therapeutic approaches to patients with benign colonic diseases could include the elimination of biofilms and restoration of normal commensal bacteria populations. However, these studies will be extremely difficult unless investigators can develop noninvasive methods for measuring and identifying biofilms. These methods that might include the measurement of quorum sensing molecules, measurement of bile acids, and identification of bacteria uniquely associated with biofilms in the colon.

## 1. Introduction

Biofilms make important contributions to the environment, industry, and health [1,2]. The composition of biofilms includes bacteria adherent to surfaces embedded in an extracellular matrix composed of polysaccharides, proteins, nucleic acids, and nutrients [3]. Biofilms frequently develop on devices, such as prostheses and intravascular catheters, used during patient care. These can result in infections that are difficult to manage due to increased resistance to routine antibiotics. Studies of biofilms on human surfaces are hampered by the need for simple and safe access to the surfaces. The best study location involves the development of biofilms on oral structures that can lead to gingivitis, dental plaque, and dental caries. The study of biofilms in the gastrointestinal tract is significantly more difficult due to the need for endoscopy to reach the surfaces. This review considers the role of biofilms in colonic health and colonic disease.

## 2. Biofilm Formation Based on Studies with *Pseudomonas aeruginosa*

### 2.1. Biofilm Structure

The initial step in biofilm formation involves the reversible attachment of free-floating bacteria (planktonic) or other microbes to an abiotic or biotic surface (Figure 1) [3,4]. The initial attachment may involve hydrophobic forces. This initial step or stage 1 is reversible, and the attachment may not persist. However, some bacteria undergo a phenotypic change and subsequently adhere to the surface (stage 2). This can result in a formation of more flagella and pili, which create more attachment structures. These bacteria replicate, and small microcolonies are formed (stage 3). The bacteria form an extracellular matrix network that consists of proteins, polysaccharides, DNA, and exogenous debris in the region of attachment. These microcolonies then grow into a biofilm with increased numbers of bacteria and extracellular matrix (stage 4).

Approximately 85% of the biofilm constitutes the extracellular matrix. The biofilm includes live bacteria, dead bacteria, and relatively quiescent bacteria or persisters. In addition, some biofilms contain multiple bacterial species, including bacteria that cannot independently form biofilms. Eventually, enzymes are released that break down part of the matrix and allow the dispersion of bacteria back into a free-floating (planktonic) existence (stage 5). These bacteria can then attach to other surfaces and start the formation of new biofilms.

Kragh studied biofilm formation using either single bacterial cells or aggregates as the initial source of the biofilms using in vitro culture methods and confocal microscopy to monitor growth [6]. With increasing cell density, aggregates grow better than single cells. Growth rates in aggregates depend on the location of the bacteria in the aggregate, and bacteria in the center of the aggregate have fewer progeny. Bacteria on the surface of the aggregate have much faster and better growth rates. This study demonstrates that biofilm formation is enhanced when the biofilm is initially seeded by aggregates rather than single cells. In addition, growth in an aggregate depends on cellular location, which presumably affects access to nutrients. Paula et al. analyzed the dynamics of bacterial population growth in biofilms using in vitro growth techniques and confocal microscopy [7]. They noted that bacteria settle randomly on surfaces, but only a subset of these bacteria grows. These active bacteria grow in a three-dimensional pattern by incorporating nearby bacteria into densely populated microcolonies. This clustering and microcolony formation depend on exopolysaccharides to provide a supporting structure. In addition, growth depends on environmental conditions and the availability of nutrients.

Reichhardt used confocal laser scanning microscopy to analyze *P. aeruginosa* biofilm architecture and matrix localization [8]. The use of this microscopy can help investigators determine the structure of biofilms and the localization of certain components of the biofilm. *Pseudomonas* biofilms contain high levels of Psl polysaccharides, Pel polysaccharides, alginate, and extracellular DNA. During the formation of biofilms, some of these structural elements change location. For example, Psl is initially located near the bacteria forming the initial matrix. However, as the biofilm grows, Psl moves toward the periphery. Nutrients must diffuse into the biofilm matrix, and this process depends on the diffusion coefficient of the molecule and the dimensions of the biofilm matrix [9]. In addition, some bacteria form microchannels in the biofilm which allows the diffusion of molecules [10]. Studies with antibiotics provide another method of monitoring molecular diffusion into the matrix and are relevant to understanding that antibiotic resistance is associated with biofilms. For example, ciprofloxacin rapidly penetrates into biofilms during static incubation. However, tobramycin accumulates on the periphery of the biofilm aggregate and is tightly adherent to various structures since it is not removed by rinsing the biofilm.

### 2.2. Bacterial Attachment to Surfaces

Studies with the bacterium *P. aeruginosa* provide important information about the attachment process (Figure 2). One important consideration is how bacteria sense a surface [11]. Following irreversible attachment, cells start to multiply and produce the matrix. In Gram-negative bacteria, the key intracellular signaling molecule is cyclic diguanylate. With high levels of cyclic diguanylate, bacteria produce more of a biofilm matrix and have less flagella-mediated swimming motility. In *P. aeruginosa*, there are two distinct surface sensing mechanisms [12]. The first one involves a WSP chemosensory system [13]. In response to the surface contact, the WSP system stimulates the production of cyclic diguanylate. The initial event is uncertain; it probably involves distortion of the cell membrane, which then activates membrane proteins. A second surface sensing mechanism involves type 4 pili. Upon contact with the surface, the methyl-accepting chemotaxis protein Pil J transduces a signal to the protein CyaB, stimulating its activity [14]. This protein increases cellular levels of cyclic adenosine monophosphate (AMP), which increases type 4 pili production and twitching motility. PilY1 has a von Willebrand motif that might have a mechanosensory function [15]. However, in both systems, it is unclear what the exact signal is that is triggered by contact. It is thought that, through successive surface interaction or attachment and detachment, cells become surface adapted [16]. This results in a progressive increase in cellular cyclic AMP and a gradual increase in type 4 pili.

Pili support several essential physiological processes in the gut microbes [17,18,19]. Type 4 pili are involved in the attachment of bacteria to other cells, such as bacteria and epithelial cells, as well as to food and fiber in the gut lumen. Attachment to host cells influences signaling and gene expression and increases the production of cyclic AMP and cyclic diguanylate. Pili are involved in adhesion, biofilm formation, motility, and molecule exchange, including DNA uptake and protein secretions. These activities increase the residence time of bacteria in the biofilm, allow nutrient exchange, promote horizontal gene transfer, increase survival, and provide protection from physical stressors. DNA exchange between bacteria can occur through three processes, including conjugation, transduction, and natural transformation. This transfer of nucleic acids is mediated by type 4 pili and other filamentous structures. Type 4 pili also allow the secretion of proteins that are essential for the bacterial colony.

Type 4 pili are extracellular filaments 6–8 nm wide and several micrometers long, composed of Pil A subunits that are assembled in a helical manner [12]. The Pil A leader sequence is approximately 10 amino acids; the mature Pil A is 140 to 160 amino acids. After the peptide sequence is oriented correctly in the inner membrane, the leader sequence is cleaved off, and the Pil A peptide is then incorporated into the growing pilus structure. One ATPase attaches the Pil A to the growing pilus; a second ATPase detaches the Pil A from the pilus as it is retracted. Chang and colleagues published a detailed study on the formation and architecture of type 4 pili [20].

**Figure 2 ijms-23-14259-f002:**
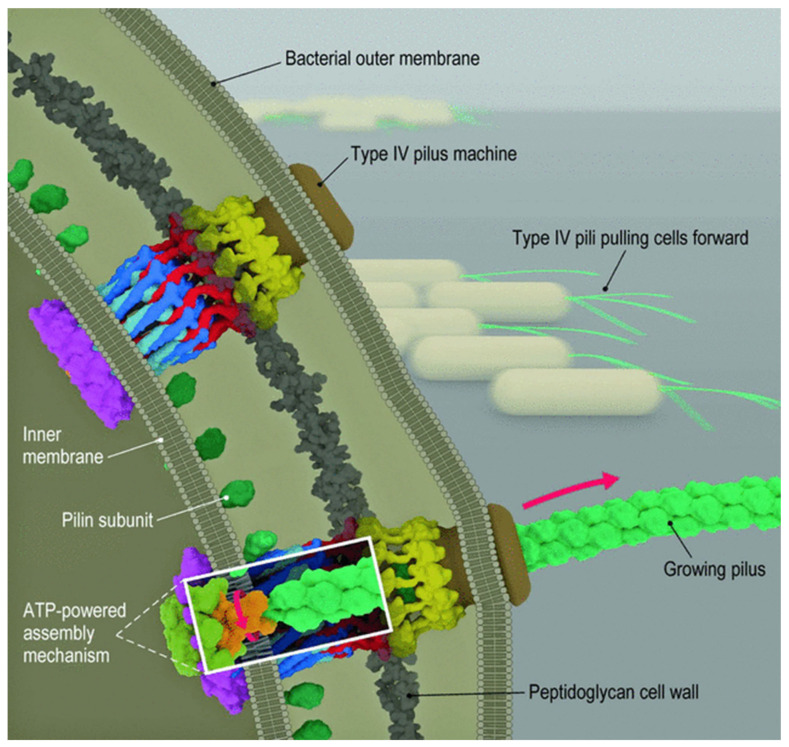
Structure and function of type 4 pili. The addition of peptides to the growing pilus requires ATP for energy. After attachment of the pilus to a surface, retraction of the pilus requires energy provided by ATP. This results in the movement of the bacterium [20].

Retraction of the pilus generates mechanical force and results in motility. Beaussart et al. studied adhesion and mechanical stress of type 4 pili of *P. aeruginosa* using atomic force microscopy [21]. They demonstrated that these bacteria adhered firmly to hydrophobic surfaces in a time-dependent manner but only weakly adhered to hydrophilic surfaces. PilY1 is a possible pilus-associated adhesin required for colonization of epithelial cells. In this study, the pili had an average length of 1.1 ± 0.3 µm and an average diameter of 4.2 ± 1.1 nm. Multiple force–distance curves demonstrated that these pili had an average adhesion force of 50–250 pN and an average rupture length of 50 to 2000 nm. These stretching experiments resulted in the development of a constant-force region that likely represents uncoiling or unfolding of their helical quaternary structure. Other studies suggested that they can behave as nanosprings. Increasing the adhesion time increases the mean adhesion force to 3000 pN. Studies have also suggested that Pil Y1 may have no direct role in these measurements. These investigators concluded that the adhesion of *Pseudomonas* to hydrophobic surfaces involves both strong cohesive interactions based on time-dependent interactions over short distances with cell surface constituents, such as membrane proteins and other surface structures, and weaker interactions involving extension and force-induced configuration changes in the type 4 pili over longer distances [21]. *Pseudomonas* type 4 pili also bind to living pneumocytes [21].

In summary, biofilm formation requires the adherence of bacteria to surface structures, the production of an extracellular matrix, growth of bacteria, and communication between bacteria through the production of quorum sensing of molecules. Metabolic activities in the biofilm include alterations in gene transcription, the uptake of exogenous DNA resulting in new biosynthetic activity and possibly antibiotic resistance, and protein synthesis and secretion. The structure of the biofilm provides a “protective shelter” which increases antibiotic resistance and reduces the interaction of phagocytes with the bacteria. Studies using *Pseudomonas* spp. have made significant contributions to the understanding of biofilm formation. An anaerobic colonic microbe, *Bacteroides fragilis*, is discussed in a later section of this review.

## 3. Quorum Sensing

For decades, conventional thinking has maintained that bacteria could only use simple biological processes since they are unicellular organisms. However, studies over the last 50 years have shown that these bacteria can “communicate” with each other and act collectively, thereby demonstrating a wide variety of complex “social” behaviors [22,23,24,25,26]. It is now evident that these social interactions have significant effects on bacterial behavior and structure of their polymicrobial communities. Their coordinated behaviors include the production of virulence factors [27,28], sporulation [29,30], control of secondary metabolite production [31], bioluminescence [32,33], genetic competence [34,35], and biofilm formation [36,37,38,39]. To successfully carry out these processes, a complex population-wide coordination of individual cells is required, and, in order for bacteria to orchestrate these collective behaviors, they use a cell-to-cell communication process known as quorum sensing (QS) [25,38,40,41].

Quorum sensing is a bacterial process that involves the synthesis, secretion, detection, and population-wide responses to extracellular signaling molecules referred to as autoinducers (Ais). As the bacterial population density increases, these Ais accumulate in the microenvironment, and this allows bacteria to detect changes in their cell densities and initiate group-wide gene expression changes [42]. Although there are many regulatory components and molecular mechanisms involved in QS, they all depend on three mechanisms. First, each member of the community produces AIs, which, when present at concentrations below detection threshold as a result of low cell density, diffuse away. However, at higher cell densities, the production of AIs by these cells within the community leads to cumulatively high concentrations of AIs with subsequent detection and response [43]. Second, autoinducers produced are detected by specific receptors located in the bacterial membrane and cytoplasm. Third, the detection of AIs leads to activation of more AI production and activation of genes required for cooperative behaviors [44,45]. These feed-forward autoinduction loops are thought to promote synchrony within the population [41].

The QS systems employed are different depending on the type of bacteria. Gram-positive QS bacteria use oligopeptides called autoinducing peptides (AIPs) as their signaling molecules. After these AIPs are produced, processed, and secreted in high concentrations, they bind to their partner membrane-bound two-component histidine kinase receptors [46,47], thereby activating their kinase activity. The kinase autophosphorylates, leading to the phosphorylation of a cognate cytoplasmic response regulator and subsequent activation of the transcription of the genes in quorum sensing [42]. In some cases, AIPs can be transported back into the bacterial cytoplasm and then interact with and modulate the activity of various transcription factors resulting in gene expression changes [42]. Gram-negative QS bacteria primarily use small molecules, acyl-homoserine lactones, as autoinducers [48] or autoinducers produced from S-adenosylmethionine [40]. At high cell density, autoinducers produced in Gram-negative bacteria diffuse across the membrane and bind to cytoplasmic transcription factors or to transmembrane two-component histidine sensor kinases. In both scenarios, AI–receptor complexes regulate the expression of QS-dependent target genes [40]. For example, *P. aeruginosa* uses AIs, such as N-3-oxo-dodecanoyl-Lhomoserine lactone (3O-C12-HSL) and N-butyryl-L-homoserine lactone (C4-HSL) from the AHL family, to control hundreds of genes involved in regulation of virulence [49]. Studies have shown that, in wildtype *P. aeruginosa*, 3O-C12-HSL-mediated cell-volume increases are associated with upregulation of aquaporin 9, a marker of chronic inflammation in inflammatory bowel disease (IBD) [50]. Using liquid chromatography and mass spectrometry, AHLs have been detected in feces from both healthy individuals and patients with IBD with flares or in remission. Fecal 3-oxo-C21 levels (a major QS molecule from gut microbiota [51]) were positively correlated with higher counts of intestinal bacteria, such as *Faecalibacterium prausnitzii*, *Clostridium coccoides*, and *Clostridium leptum* [51], and high levels of toxin-induced QS signaling peptides have also been reported in feces of *Clostridium difficile* diarrheal patients when compared to *C. difficile*-negative diarrhea patients [52].

Information encoded in QS autoinducers are characteristically integrated into the control of gene expression, thereby enabling intra-species, intra-genera, and/or cross-species communication, as well as bacterial species within the microbiota (Figure 3) [40]. Many traits in a given bacterium can be under quorum sensing control. The increased production of autoinducers when identical autoinducers are detected generates a feed-forward regulatory loop, while other features, including feedback loops and small regulatory mRNAs, small nucleoid protein Fis, and alternate sigma factor RpoN, optimize the integration of the information encoded by the AIs, thereby providing ideal QS dynamics [53,54,55].

The current understanding of the mechanisms of quorum sensing have developed largely from studies using pure in vitro bacterial cultures. Although these studies have definitely provided a better understanding of the molecular mechanisms used in quorum sensing, in nature, bacteria often exist in communities of mixed species within a given microenvironment and in which fluctuations may also occur [56]. Therefore, more research effort is needed to study communication between these microbes and their behavior in settings that more closely resemble what occurs in nature and in humans.

The detection of QS in stool specimens could provide evidence of the presence of biofilms in the colon, and the concentration of QS might indicate the amount of biofilm present. These possibilities need study in patients with a variety of colonic diseases, both before and during treatment, and in normal control subjects [51]. In addition, the QS levels need correlation with the predominant stool bacteria using molecular methods for identification.

**Figure 3 ijms-23-14259-f003:**
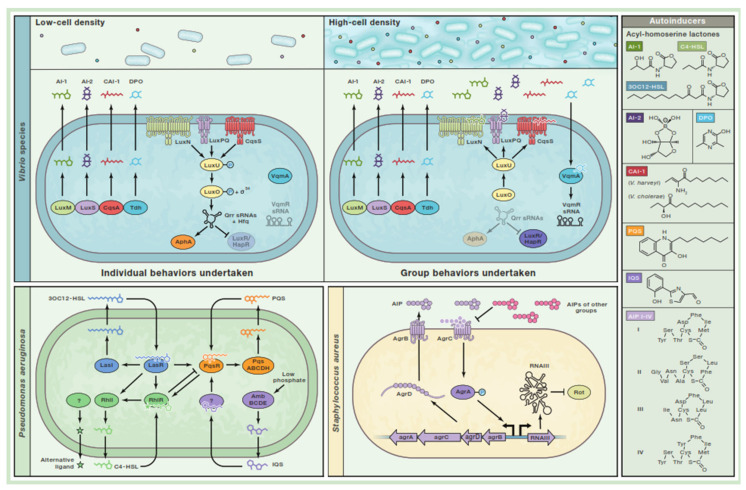
Bacterial quorum sensing. In *Vibrio* species, when autoinducer (AI) levels are low due to low cell density, the AI cognate receptors activate a phosphorylation cascade that leads to the activation of transcription factor AphA, a mediator of individual cell behaviors. In contrast, at high cell density (HCD), LuxM, LuxS, CqsA, and Tdh synthases produce high levels of AI-1 (3OHC4-homoserine lactone (HSL)), AI-2 ((2S,4S)-2-methyl-2,3,3,4-tetrahydroxytetrahydrofuran-borate), CAI-1 ((Z)-3-aminoundec-2-en-4-one), and DPO (3,5-dimethylpyrazin-2-ol), respectively. The AIs bind their respective receptors leading to the abrogation of AphA activity and LuXR or HapR activation. Under these conditions, *Vibrio* species initiate group behaviors. *P. aeruginosa* uses four pathways that employ cytoplasmic AI receptors acting as transcription factors when bound by AIs. LasI/LasR and RhII/RhIR produce and detect 3OC12-HSL and C4-HSL, respectively, while PQS (2-heptyl-3-hydroxyl-4-quinolone) produced by PqsABCDH is bound by the PqsR receptor. The fourth system is dependent on the production of IQS (2-(2-hydroxyphenyl)-thiazole-4-carbaldehyde) by AmbBCDE and subsequent activation of the Pseudomonas quinolone signal (PQS) system by IQS via an unknown receptor. At HCD, a precursor peptide (AgrD) is processed by AgrB into inducing peptide (AIP), which, when exported, activates AgrC, an autokinase receptor, which then phosphorylates AgrA. Phosphorylated AgrA induces the production of a regulatory RNA that controls group behaviors. Adapted from Eickhoff et al. [57] with permission.

## 4. Detection of Biofilms

Biofilms are highly structured communities of microbes that are adherent to biotic surfaces, such as host cells, or abiotic surfaces, such as medical devices, and they are surrounded by self-produced extracellular polymeric matrices [58]. Infections caused by biofilm-forming microbes remain important clinical problems due to their persistence, resistance to many antimicrobials, and evasion of host defenses [59]. Furthermore, increased morbidity and mortality rates and hospital costs are associated with biofilm formation, particularly around devices, such as catheters and implants [60]. Even after decades of research, the diagnosis of biofilm-related infections remains difficult since patients infected with biofilms do not have unique symptoms or presentations. Although there are several standardized methods for detecting biofilms in research laboratories, no such protocol exists in clinical practice. In addition, new antimicrobials under development are being tested on planktonic bacteria, which may have antibiotic sensitivities different from bacteria in biofilms [61].

### 4.1. In Vitro Techniques for Biofilm Detection

#### 4.1.1. Direct Observation

The complexity and dynamics of biofilms can be observed through imaging technologies, such as light microscopy, electron microscopy, and confocal microscopy. These techniques allow researchers to identify biofilms and visualize their 3D structures [62].

##### Light Microscope

Light microscopy provides the most convenient and fastest method to study the morphology of microorganisms adherent to surfaces and to semi-quantitatively estimate the number of microorganisms attached to the surface (e.g., absent, rare, present, or abundant). Although light microscopy does not allow for 3D visualization of biofilm since it requires clear, transparent, and planar surfaces, both Gram-positive and Gram-negative bacteria adherent to abiotic surfaces have been observed by light microscopy, which allows for comparison of the morphology of sessile forms and planktonic forms of the microorganism.

##### Fluorescence In Situ Hybridization (FISH)

Specific microbes present in a heterogeneous biofilm community can be identified through the use of probes (short fluorescence-labeled oligonucleotides) that bind specifically to the ribosomal RNA of target organisms [63]. The growth rate of organisms in the biofilm can be measured by FISH since the number of ribosomes in a given microbe is directly proportional to the growth activity of that organism. The fluorescence in situ hybridization method has the advantage of direct sample analysis without prior treatment and easy identification of microbial aggregates [64]. Using FISH and confocal laser scanning microscopy (CLSM), Bernardi and colleagues were able to identify the bacteria responsible for biofilms observed in the tongue dorsum of patients with halitosis [65]. This method has also been used to identify biofilm-forming bacteria in other clinical specimen, such as vaginal samples [66].

##### Electron and Confocal

Electron microscopy provides a platform on which microbial aggregates can be visualized following sonication or FISH (Figure 4). The most commonly used tools are scanning electron microscopy (SEM), transmission electron microscopy (TEM), and confocal laser scanning microscopy (CLSM) [67]. Due to its high-powered resolution, this technology provides direct detection and visualization of biofilms and detailed analysis of their structures [68,69]. The use of SEM and TEM requires the samples to be fixed, dehydrated, and stained, processes that could alter the morphology and structure of the biofilm being studied [67,70]. One of the main advantages of the CLSM is that it provides a complete visualization of the 3D biofilm architecture, thereby allowing identification of macromolecules and components within the biofilm and their extracellular matrix [67].

Although in vitro analysis of biofilms has undoubtedly contributed to the understanding of biofilm biology, with many of the existing technologies, biofilms are analyzed in an environment dissimilar to where they are formed [69]. To circumvent this challenge, the use of multiphoton laser scanning microscopy has allowed for direct detection and visualization of biofilm in situ, making it possible to study and analyze biofilms at their sites of formation, such as implanted medical devices [73].

#### 4.1.2. Indirect Observation

##### Roll Plate Method

The roll plate method is a semiquantitative method for detecting possible microbial colonization on indwelling device-associated infections on the outer surface of cylindrical devices, such as catheters and vascular grafts [74]. This method is performed by rolling the medical device tip back and forth on the surface of agar plate, followed by incubation and counting the number of colony-forming units. This method has been used in the diagnosis of catheter colonization and bacteremia in hospitalized patients [75].

##### Congo Red Agar Test

The Congo red agar (CRA) method is a rapid, sensitive but qualitative assay that detects biofilm-producing organisms based on colony color changes inoculated on CRA. Black colonies on CRA indicate robust biofilm production; red colonies indicate the lack of biofilm production [76]. Congo red agar methods have been used in several studies to detect biofilm-producing microorganisms from clinically derived samples, such as urine, blood, pus, wound, sputum, ear swabs, and throat swabs [77,78].

##### Tube Biofilm Formation Test

Tube biofilm formation is a method used to qualitatively detect the presence of biofilm-producing microorganisms. Isolates are inoculated in plastic test tubes and incubated for 24 h at 37 °C. The sessile isolates forming biofilms on the walls of polystyrene test tubes are stained with safranin or crystal violet; nonadherent bacteria are removed by rinsing with buffered solutions. After air drying the test tubes, the presence of stained films on the bottom and walls of the tube indicates the development of biofilms [77,79]. The amount of biofilm can be quantified by solvent extraction of the dye followed by spectrophotometry. Using the tube biofilm test, Neopane and others identified 30 *Staphylococcus aureus* biofilm-forming isolates from clinically derived samples [80].

##### Microtiter Plate Assay

The microtiter plate assay provides a quantitative method to measure biofilm production using a microtiter plate reader. It is a relatively cheap method, and several samples can be tested simultaneously [81]. To detect biofilm production, the isolates are incubated in microtiter plates followed by several washes that remove planktonic bacteria leaving only adherent cells that are then stained with crystal violet dye for biofilm visualization [82]. For quantification, the stained biofilms are solubilized, and their optical density (OD) is determined by spectrophotometer. Optical values can also provide categorization of isolates as nonproducers or weak, moderate, or strong biofilm producers [83].

##### Polymerase Chain Reaction

Polymerase chain reactions (PCRs) are used to identify specific pathogens by amplifying species-specific nucleic acid sequences and to detect biofilm-associated genes using specific primers, even directly in uncultured clinical samples after sonication [84]. Several gene amplification-based technologies, such as real-time PCR, multiplex PCR, and reverse transcriptase PCR, have been used to identify biofilm-forming microbes from clinical samples, such as urine, blood, sputum, cerebrospinal fluid, pleural fluid, and wound samples [85]. Specific genes involved in bacterial adhesion, a necessary step in biofilm formation, were also identified in different strains of *Listeria monocytogenes* using real-time PCR [86].

### 4.2. In Vivo Techniques for Detection of Biofilms

Studies using mammalian models for the in vivo study of biofilm biology have been limited due to logistical and ethical challenges involved in carrying out such studies. In the past two decades, to circumvent these practical difficulties, several non-mammalian models have been proposed and used, including plant models, such as thale cress (*Arabidopsis thaliana*) and duckweed (*Lemma minor*), in which the virulence of pathogenic *Staphylococcus aureus* and *P. aeruginosa* was successfully correlated with biofilm formation [87]. Other complex models, such as *Drosophila melanogaster* [88,89], *Caenorhabditis elegans* [90], and zebrafish [91], have also been studied.

#### Low-Coherence Interferometry and Optical Coherence Tomography

Due to its high resolution and imaging capabilities, low-coherence interferometry (LCI) and optical coherence tomography (OCT) [92] can detect and quantify of biofilms in patients’ middle ears. This is a noninvasive procedure with high-resolution depth-ranging and imaging capabilities. With the use of LCI/OCT technology, Nguyen and colleagues detected biofilms in the middle ears of rats [93,94]. Using the same technology in a clinical study, these authors detected the presence of biofilms in the middle ears of adult patients with chronic otitis media [95]. In another clinical study using catheter-based optical coherence tomography, endotracheal tube biofilms were detected and quantified in intubated critical care patients [96].

Despite the research progress made over the last three decades, microbes that produce biofilms remain a major public health concern because biofilms are associated with increased antimicrobial resistance and a worse prognosis for patients. The clinical diagnosis of biofilms remains challenging as current detection methods are more applicable in research settings. Therefore, more studies are needed to develop efficient and reliable tools for rapid biofilm identification and improve chances for successful infection control in clinical practice. Potential approaches include the detection of bacteria frequently associated with biofilms and the measurement of key metabolites associated with biofilms in stool samples.

## 5. Microbiome and Biofilms in the Colon

The term microbiome generally refers to the entire habitat, which includes the microorganisms, their genomes, and the surrounding environment. In humans, the gastrointestinal tract is the site most densely populated with microorganisms. Previous studies on the gut microbiome showed that the gut is the habitat of Gram-positive *Firmicutes* and Gram-negative *Bacteroidetes*; Actinomycetes (Gram+ anaerobes), methanogens (methane producing prokaryotic Archaea), and fungi are present in lower quantities. Most *Firmicutes* are in the genus clostridia and can produce butyrate. Several *Proteobacteria* and *Actinomycetes* were also identified, and *Bifidobacteria* (subgroup of *Actinomycetes*) represented 5% of the microbiota. Eukaryotic microbes in the human gut consist of *Blastocystis* sp. (a single-celled parasite) and fungi in the phylum Ascomycota (*Ascomycetes)* and the phylum Basidiomycota (*Basidiomycetes*). The majority of the Ascomycetes belong to the genera *Candida albicans*, *C. glabrata*, *Penicillium italicum*, *P. glabrum*, *P. sacculum*, *P. verruculosum*, *Saccharomyces cerevisiae*, *S. cariocanus*, and *S. bayanus* [97].

The colon has important environmental features that have a strong effect on the bacterial population. The oxygen content in the colon is extremely low, and ≥99% of the bacterial species in the colon are anaerobic [98]. The physiologic activity of these bacteria has important effects on the health of the colonic epithelium; these activities include both nutritional support and immune system maturation with protection from pathogenic bacteria. Consequently, the formation of commensal bacterial communities on the mucus layer of the epithelium and on biofilms becomes essential to maintain normal physiologic functions. *Bacteroides fragilis* is an important bacterium in the colon and has both beneficial effects and potentially pathogenic effects. This bacterium is an anaerobic, Gram-negative, pleomorphic, mostly rod-shaped bacterium that represents approximately 25% of the fecal microbiota [98]. It can metabolize biopolymers, polysaccharides, and glycoproteins into smaller molecules, which are subsequently used by epithelial cells and/or other microbes. Pumbwe et al. studied the effect of bile salts on *B. fragilis* in in vitro assays [99] These studies demonstrated that bile salts increased bacterial co-aggregation through the production of fimbria-like appendages and outer membrane vesicles. The salts also increased the formation of efflux pumps, and this increased resistance to antibiotics. Lastly, bile salts increased adherence to intestinal epithelial cells and stimulated biofilm formation. Therefore, this study indicates that bile salts at the right concentration have important effects on *B. fragilis* and probably contribute to its survival in the normal colon and its virulence in the inflamed colon. These investigators also studied the effect of quorum sensing molecules on biofilm formation by *B. fragilis* [100]. Homoserine lactone and cell-free supernatants from cultures of other bacteria were added to *B. fragilis* in in vitro growth assays. These compounds slowed growth, increased efflux pump gene expression, increased resistance to antibiotics, and increased biofilm formation. Consequently, *B. fragilis* in mixed communities of bacteria probably responds to the release of quorum sensing molecules from the other bacteria, and this potentially promotes their growth and stability, increases antibiotic resistance, and increases biofilm formation. *B. fragilis* isolated from different body sites, including blood, abscesses, and stool, have different phenotypic characteristics [101]. For example, isolates from the stool were tolerant to stress secondary to increased concentrations of sodium chloride and bile salts. In addition, isolates from the stool had surface vesicles budding from the outer membrane. Consequently, this study indicates that bacteria adapt to the microenvironment, and this affects gene expression and phenotypic characteristics of the bacterium. Overall, the complex bacterial communities in the colon have important beneficial effects and potentially adverse effects.

The mucosa of the colon consists of a single layer of epithelial cells that are covered by a layer of mucus ranging from 50–800 µm thick; the inner layer (30 µm) is dense and free of bacteria [102]. The outer loose layer of mucus provides a scaffold for commensal bacteria. Other cells include enteroendocrine cells, goblet cells that secrete mucins, and Paneth cells that secrete antimicrobial peptides. Dendritic cells, B cells secreting IgA, and regulatory T cells are located beneath the epithelial cells. Commensal bacteria are associated with the outer layer of mucus, in biofilms on particulates in the colon, and in the stool as solitary bacteria. Complex interactions between the epithelial cells and commensal bacteria are required to maintain the health of both cellular populations [103]. Important issues include the metabolic activity of the epithelial cells and bacteria, nutritional support by bacteria for the epithelial cells, and beneficial host defenses, which prevent bacterial invasion of mucosa and control the type and number of commensal bacteria. In addition, the complex three-dimensional structure of a biofilm has effects on host defenses, which may limit the clearance of pathogenic bacteria, if present, from these regions.

The colon has an extremely large surface area (250–400 m^2^) and a large collection of bacteria that includes 500–1000 different species, mostly anaerobes (~99%) [17]. It has unique features relevant to the attachment of bacteria to epithelial surfaces and to biofilms. Colonic mucus grows at a speed of 240 µm/h. Epithelial cells are shed and replaced at a rate of 100,000,000 to 300,000,000/h in the colon [3]. Both these processes result in a relatively rapid turnover of cells and acellular material on the colonic epithelium, potentially affecting the stability of bacteria on surfaces and on any associated biofilms. In addition, there is peristalsis, which moves fluid and debris across these surfaces. These factors might favor the formation of biofilms in only certain locations, such as the appendix in the normal colon. Changes in commensal bacteria and the development of biofilms could provide an early warning signal for changes in colonic epithelial health and function.

The normal colon is usually covered with a mucus layer with scattered adherent bacteria. Swidsinski et al. studied the mucus barrier in healthy colons and inflamed colons of 20 control patients, 20 patients with self-limited colitis, and 20 patients with UC [103]. The biopsies were performed in ‘purged colons’ (defined as colons prepared with enemas), and the bacteria were studied using the FISH technique. This study showed that biopsy specimens from the healthy subjects had approximately 60% of the epithelial surface covered with mucus. Bacteria were detected in mucus in about 20% of the biopsy specimens and covered 1–10% of the mucus surface [103]. In patients with self-limiting colitis and UC, the mucus layer on the colon surface was thinner, more so in patients with UC than self-limiting colitis. The mucosal surface in these patients also had more mucus-penetrating and adherent bacteria than the control patients. Therefore, this study suggests that inflammation in both UC and self-limiting colitis allows increased bacterial penetration and invasion of the mucosa compared to healthy control subjects.

Another study using colon biopsies found that approximately 13% (15 out of 120) of healthy subjects undergoing routine screening colonoscopy had a thin layer of bacterial biofilms on the colon surface with an average density of 10^8^ bacteria/mL. This biofilm was found throughout the colon [104]. Baumgartner studied biofilms from colonic biopsies and determined that the prevalence of biofilm was 19% (212 out of 1112) and that only 6% of the biofilms were found in healthy control patients [105]. This study demonstrated that healthy controls had a lower biofilm prevalence, which supports the hypothesis that biofilm formation is linked to the pathological state of the microbiome [105]. Colon biopsies from healthy humans are covered by thin biofilms, consisting mostly of Bacteroidetes (a phylum with three classes of Gram-negative rods), Lachnospiraceae (a family of obligate anaerobes which ferment short-chain fatty acids), and Enterobacteriaceae (a family of Gram-negative rods which ferment sugars to lactate). Some bacterial species that cannot form biofilms by themselves are also present in mixed biofilm communities with other strong biofilm-forming species. Commensal bacteria, including *Parvimonas* (a genus of Gram-positive anaerobes), *Peptostreptococcus* (a genus of Gram-positive anaerobes), and *Prevotella* (a genus of Gram-negative bacteria), are often detected in intestinal biofilms [19,106]

Macfarlane and Macfarlane studied the bacterial populations attached to food particles in stool samples [107]. These bacteria may differ from bacteria on colonic surfaces and from the free-floating bacteria. These populations were similar to free-floating bacteria collected from the stool. The biofilm populations included both living and dead bacteria. Confocal laser scanning electron microscopy demonstrated that bacteria were present both as isolated single cells and as microcolonies. Most of the bacteria isolated were anaerobic bacteria; aerobic bacteria included *Escherichia coli* and *Enterococcus faecalis*. The density of bacteria ranged from 5^10^ to 6^10^ colony-forming units per gram of wet weight stool. Bacteria in the biofilms were able to digest polysaccharides; nonadherent bacteria were able to better digest oligosaccharides. This study demonstrates that bacteria can form biofilms on particulate debris in stool and have different metabolic activity in comparison to free-floating bacteria.

### 5.1. Metabolic Activity in Biofilms

Nutritional support for the intestinal epithelial cells reflects the unique features in the colonic environment [17]. The oxygen content of the colon is extremely low, and this dictates the need for anaerobic metabolism. Colonic contents include undigested fibers and polysaccharides. Bacteria in the biofilms digest these fibers and produce short-chain fatty acids, including butyric, propionic acid, and acetic acid. In turn, the butyric acid is used by mitochondria in epithelial cells to produce ATP.

### 5.2. Host Defenses at the Colonic Surface

The colon is constantly exposed to a large number of microbes and innumerable antigens [17,19,108]. It must maintain a fine balance between the protection from pathogenic microbes without the development of deleterious immune responses that could cause tissue injury. The anatomic structure of the colonic mucosa and a finely regulated immune response provide this protection. Colonic health also depends on the number and types of commensal bacteria in the colon on epithelial surfaces and in the lumen, as well as possibly in colonic biofilms. The dense mucin layer on the intestinal epithelial cells prevents bacterial attachment and invasion into the epithelium. B cells in the subepithelial layer produce secretory IgA that binds to bacteria and prevents adherence to surface structures. Immune exclusion prevents bacterial movement or translocation across the epithelial barrier, which is prevented by the mucus layer and by secretory IgA. Specialized Paneth cells in the epithelium produce antimicrobial products that have bactericidal activity. Lastly, a finely regulated adaptive immune response prevents bacterial infection but limits associated inflammation.

Biofilms in patients with intestinal disease may undergo significant alteration in composition. These alterations can include a dense bacterial collection on the biofilm associated with a decrease in bacterial diversity and expansion of certain pathogenic bacteria. One such bacterium is the adherent and invasive *E. coli.* This *E. coli* strain can invade epithelial cells, can survive and replicate in epithelial cells and macrophages, can cause colitis, and can induce specific T-cell responses. Curli fibers, which are bacterial amyloid fibers, modulate immune responses and host–microbial interactions on the mucosa and possibly serve as a protective bacterial factor that limits infections [109]. Bacterial cellulose production also modulates host microbial interactions and disease production, and Pili 1 interacts with receptors on epithelial surface cells. These two bacterial molecules likely increase inflammation [109]. However, it is difficult to determine whether or not the interaction with bacteria and bacterial extracellular molecules have beneficial or harmful effects on epithelial surfaces, and it may depend in part on the location of the interaction, namely luminal surfaces versus basal lateral surfaces.

### 5.3. Bacterial Protection in Biofilms [110]

A mucin layer on the intestinal epithelial cell surface and biofilms create safe environments for commensal bacteria. Phagocytic cells, such as neutrophils, may not be able to enter the biofilm matrix and cannot ingest interior bacteria. Some antibiotics adhere to the components of the matrix, and this prevents an antibiotic effect on the bacteria. Gene transfer processes in the bacterial community can transfer antibiotic resistance genes to antibiotics susceptible bacteria and prevent their killing.

### 5.4. Other Factors Relevant to Biofilm Formation and Stability in the Colon

Biofilm formation represents a complex process which depends on the bacterial population, the presence of surface structures which provide the physical basis for biofilm, and exogenous material which potentially contributes to the extracellular matrix and the nutrition of the bacteria. Several factors potentially influence biofilm stability and/or turnover. Biofilm formation of the colon is likely more complicated than biofilm formation on medical devices and possibly other tissue sites, such as the gingiva. In the colon, biofilm and epithelial health depends on the nutrients in the colonic lumen [98]. In particular, the digestion of polysaccharides by colonic bacteria provides small molecules, which in turn are used for nutritional support of the bacteria and the colonic epithelium. Anderson and coworkers studied the effect of long-term fluctuations of diet on the oral biofilm [111]. They demonstrated that elevated sucrose consumption favored the development of non-mutans *Streptococci* and that frequent milk and yogurt intake decreased the abundance of these microbes. Although the colon was not directly studied in this project, it seems reasonable to think that alterations in the diet will affect the colonic microbiota. Akimbekov et al. reviewed vitamin D and the host gut microbiome [112]. They reported that vitamin D has both anti-inflammatory and immune modulating effects in the gastrointestinal tract. These effects will influence the human microbiome. Consequently, changes in vitamin D intake and possibly sun exposure may have important effects on colonic biofilm formation.

Exogenous factors also affect the gut microbiota. Thomas and coworkers studied oral biofilms in 22 subjects [113]. They used the 16 S rRNA gene to identify the microbial species and found a significant decrease in species richness in only smokers and in smoker/alcohol drinkers when compared to control subjects. These changes made the biofilms a more homogeneous microenvironment, which could contribute to the development of oral diseases. Several studies have indicated that bacteria in the oral cavity are also found in the colonic microbiota [114,115]. Consequently, it seems possible that tobacco and alcohol use affects the oral microbiome and, in turn, has the potential to affect the colonic microbiota. Changes in the use of these two products would potentially make changes in the stability of the biofilm in the colon and make serial studies more difficult. Ramirez et al. described antibiotics is a “major disrupter of the gut microbiota” [116]. Antibiotics can reduce species diversity, reduce metabolic activity, and increase the selection of antibiotic resistant organisms. These changes may be particularly important with antibiotic-associated diarrhea and recurrent *Clostridium difficile* infections. Osmotic laxatives also potentially alter the bacterial population in the colon. Tomkovich and coworkers studied the effect of an osmotic laxative in a murine model of *C. difficile* colitis [117]. Their results indicate that a laxative decreased resistance to colonization by *C. difficile* and prevented clearance in mice already colonized with *C. difficile*. Consequently, multiple factors potentially influence the development and stability of biofilms in the human colon; in particular, diet, alcohol, cigarette smoke, antibiotics, and laxatives may have important effects which complicate clinical studies.

### 5.5. Colonic Biofilms and Disease

Alterations in biofilms are associated with colonic diseases. Biofilms can cause disruption of colonic epithelial cells, which results in increased permeability in the gut, causes loss of function of the intestinal barrier, and generates intestinal dysbiosis [19]. The formation of biofilm in the gut epithelium can also disrupt the protective mucous layer and allow increased contact with microbes resulting in reduced clearance of infection [118]. Biofilm-related diseases typically arise from chronic persistent infections that cannot be eliminated by the immune system and may accelerate collateral tissue damage [119]. Multiple studies on colonic biofilm and associated diseases have shown that biofilms are linked to pathogenesis of colorectal cancer, familial adenomatous polyposis, inflammatory bowel disease (IBD), irritable bowel syndrome (IBS), and recurrent *Clostridium difficile* infection [19,105,120]. However, invasive bacterial biofilms have been identified on normal colon mucosa in approximately 13% of the healthy subjects [121], and the significance of these biofilms is uncertain. The next sections consider biofilm formation in patients with benign colonic diseases.

## 6. Inflammatory Bowel Disease (IBD)

### 6.1. Alteration of Gut Microbiome in IBD

Scanlan studied human methanogen diversity in healthy patients and patients with IBD using fecal DNA extracts and demonstrated an overall decrease in bacterial diversity and in methanogen diversity in IBD patients [122]. However, an increase in fungal diversity was noted [123]. Different studies have identified pathogenic microbes in the gastrointestinal tract of IBD patients [97]. *Saccharomyces cerevisiae*, *Candida albicans, Listeria monocytogenes*, *Mycobacterium avium* subsp. paratuberculosis, *Chlamydia pneumonia*, and adherent, invasive *E. coli* (AIEC) have been reported as potentially pathogenic microbes in the progression of Crohn’s disease (CD). The bacterial population in patients with CD and ulcerative colitis (UC) has an increase in the number of *Bacteroides*, *Peptostreptococcus*, and *Eubacteria*; there is also an increase in virulent *E. coli*, enterotoxigenic *B. fragilis,* and *P. aeruginosa* [110]. There is a decrease in the number of *Bifidobacteria* and *Faecalibacterium prausnitzii.* In UC patients, the presence of facultative anaerobic bacteria is increased [97].

### 6.2. Microbiome–Host Immune System Interactions and the Pathogenesis in IBD

One hypothesis about the pathogenesis of IBD maintains that the disease may not be caused by any specific microbial infection; instead, it develops secondary to a change in the overall composition of microbes in the gut [97]. Another hypothesis states IBD may be caused by misrecognition of normal, commensal bacteria as pathogens, leading to immune responses and inflammation. This change in gut flora could modify gene expression since microbes regulate their gene expression in response to changing microbial cell population and its density through cell to cell communication, i.e., quorum sensing, as discussed in earlier sections of this review [97].

During changes in microbiome composition or genetic biosis in inflammatory bowel diseases, the mucosal barrier is disrupted, and the epithelia-adherent biofilm is increased [110]. In patients with UC, the goblet cells are depleted, and the mucus layer formed is very thin [97].

### 6.3. Biofilms and IBD

The impact of biofilm in the pathogenesis of IBD is not well understood. Adhesive, invasive *E. coli* (AIEC) is an invasive bacterium and has been isolated from tissue samples; it has a nonspecific distribution pattern in the colon [124]. AIEC can also be found in healthy subjects despite the virulence genes. Martinez-Medina studied the molecular diversity of *E. coli*, specifically AIEC, in 20 CD patients and 28 healthy controls. The colonic biopsies were performed, and AIEC was identified using the PCR technique [125]. The prevalence of AIEC in healthy subjects was less than in patients with IBD and ranged from 0% to 16% of colonic samples. It was more abundant in the ileum than in the colon in healthy controls. In contrast, AIEC was found in 55% of ileum specimens and 50% of the colon specimens of CD patients [125]. Patients with CD also have abnormal colonization of the intestine by AIEC identified mucosa. These bacteria adhere to the epithelium and colonize and survive inside macrophages. These strains have significantly higher biofilm formation and greater adherence and invasion indices than other colonic *E. coli* isolates [124]. While only 7–10% of UC patients were found to have AIEC, studies by Sasaki et al. suggested AIEC from patients with UC were less invasive than the AIEC from patients with CD, even though the immune responses with TNF-alpha production by macrophages and interleukin-8 production by epithelial cells, as well as the decreases in epithelial barrier function, were similar [126]. However, studies on the pathogenesis of AIEC in UC are still conflicting, and future studies are required to better understand the pathogenesis of AIEC.

Chassaing studied the association between AIEC colonization and the formation of the biofilm and the virulence of AIEC [127]. Tissues were obtained from the murine intestines and studied in vitro. This study showed that, in animals with CD, the ileum is colonized by AIEC bacteria that can adhere to and invade intestinal epithelial cells. These bacteria can also grow macrophages and form biofilms. The study demonstrated that the σ^E^-mediated pathway is activated during adhesion of AIEC strain LF82; this results in increased bacterial adherence to and invasion of intestinal epithelial cells and the formation of biofilms [127]. Furthermore, AIEC LF82 adapts to phagolysosomal stress, and this increases antibiotic resistance. In addition, intracellular LF82 produces an extra-bacterial matrix consisting of polysaccharide and curli fibers that acts as a biofilm and controls the formation of LF82 intracellular bacterial communities in phagosomes for several days post infection [128].

Swidsinski studied the composition of mucosal flora in biopsy specimens from healthy controls, as well as IBD (UC and CD), IBS, and self-limiting colitis patients, with 20 patients in each group [129]. The study showed that bacterial biofilms were strongly associated with IBD, as the mean density of mucosal biofilm in IBD was twofold higher than in patients with IBS or healthy control subjects. *Bacteroides fragilis* was responsible for more than 60% of the biofilm mass in patients with IBD but only 30% of the biofilm mass in patients with self-limited colitis and <15% of the biofilm mass in patients with IBS. The biofilm in untreated IBD patients was thick, dense, and adherent, whereas biofilm in self-limited colitis patients was more diverse, less concentrated, and loosely attached to the mucosal surface [129]. In the Baumgartner study on endoscopic visible biofilm, biofilms were present in one-third of patients with UC. Biofilms also correlated with a less diverse microbiome with overgrowth of *Ruminococcus gnavus* and *E. coli* [105].

These studies indicate that biofilm formation and function need more study in patients with IBD. Important issues include the direct detection of biofilms by endoscopists [105] and identification of the predominant bacterial species using molecular methods.

## 7. Irritable Bowel Syndrome (IBS)

### 7.1. Alterations in the Gut Microbiota

Mucosal bacteria obtained from colonic biopsies from patients were present in concentrations greater than 10^9^/mL in 65% of IBS patients compared to 35% of healthy controls [129]. In a recent study, biofilms with *E. coli* and *R. gnavus* in the terminal ileum and the ascending colon were present in 60% of IBS cases [106]. A systematic review reported that *Enterobacteriaceae*, *Lactobacillaceae*, *Bacteroides*, *Lachnospira*, and *Clostridium* were significantly increased in patients with IBS [130,131]. Swidsinski studied 20 patients with IBD, 20 patients with IBS, and 20 controls using mucosal tissue and rRNA-targeted oligonucleotide probes and reported that the mean density of the mucosal biofilm was twice as high in IBD patients as in patients with IBS or in controls [129]. Compared with IBD, *Eubacterium rectale* and *Clostridium coccoides* were found more frequently in IBS patients (>40%) than in IBD patients (<15%) [129]. Carroll studied fecal and colonic mucosal biopsy samples obtained from 10 patients who was diagnosed with IBS diarrheal type and 10 healthy controls [132]. DNA was extracted, and bacteria were identified using the PCR technique; this study demonstrated a significant reduction in aerobic bacteria in fecal samples, mainly *Lactobacillus* species, in IBS patients compared to controls [132]. Krogius-Kurikka studied the microbes in 10 patients who had IBS diarrheal type compared with 23 health controls; the fecal samples were analyzed using PCR techniques and gene sequencing [133]. The study showed that IBS diarrheal type patients have an increased number of *Proteobacteria* and *Firmicutes* and a reduction in *Actinobacteria* and *Bacteroidetes* compared with control patients. Compared with IBD, *Eubacterium rectale* and *Clostridium coccoides* were found more frequently in IBS patients (>40%) than in IBD patients (<15%) [129].

### 7.2. Microbiome–Host Immune System Interactions and Pathogenesis in IBS

There are several hypotheses potentially explaining microbiome alterations in IBS populations. Hypotheses include the loss of microbial diversity and richness, previous intestinal infection, changes in the intestinal environment, including a decrease in anaerobic bacteria, and potential inflammation and increased gastrointestinal motility, which allow the growth of non-fastidious bacteria, such as *Enterobacteriaceae* [134,135]. *Lactobacillus* can produce organic acids, including lactic acid and/or acetic acid, which are associated with bloating and abdominal pain in IBS patients, depending on patterns of fermentation [136]. Enterotoxigenic strains of *Bacteroides fragilis* can produce toxins that interrupt colonic mucosa formation and are associated with abdominal pain and diarrhea. Alteration of predominant fermenting bacteria, such as *Bacteroidales* and *Clostridiales*, might be involved in the pathophysiology of the IBS diarrheal type [131,137]. *Clostridium difficile* can increase the risk of post-infectious IBS [138]. Moreover, a significant decrease in *Firmicutes* and an increase in *Bacteroidetes* occurred in IBS diarrheal type patients [139]. In Chassard’s study, fecal samples were collected from 14 patients with IBS constipation type and 12 controls and analyzed by FISH in vitro [140]. This study demonstrated that a decrease in the number of lactate-producing and lactate-utilizing bacteria, as well as in the number of H_2_-consuming bacteria, methanogens, and reductive acetogens, was linked to the IBS constipation type group. A recent study by Baumgartner reported that bacterial biofilms were associated with gut microbial dysbiosis and increased levels of intestinal bile acids [105]. Baumgartner et al. studied colonic biopsies and fecal samples in 117 patients using molecular and microscopic analysis with 16s rRNA gene sequencing technique. These investigators reported that the prevalence of biofilms in patients with IBS, UC, and CD was 57%, 34%, and 22%, respectively, whereas the prevalence of biofilms in healthy subjects was 6%. The most common location of biofilms found during colonoscopies is the ileocecal region, independent of disease pathology [105].

### 7.3. Bacterial Biofilm and IBS

Patients who had biofilms identified during colonoscopies were found to have a decrease in bacterial richness and diversity in biopsy specimens [105]. However, the *Escherichia/Shigella* genus and *Ruminococcus gnavus* group were increased. The presence of biofilms was associated with a decrease of short-chain fatty acid-producing bacterial genera, including *Faecalibacterium*, *Coprococcus*, *Subdoligranulum*, and *Blautia*. The physical nature and size of these biofilms could reduce peristalsis and create a diffusion barrier, which could contribute to or possibly explain common symptoms, such as bile acid-induced diarrhea, bloating, and pain [105]. In a recent study with 80 patients with IBS and 65 control subjects, fecal samples had increased levels of *R. gnavus* in samples from patients with IBS and no differences in fecal microbiota composition in each IBS subtypes [141]. However, this study was limited by the lack of measurement of certain metabolites that have previously been associated with bacterial biofilms.

The association of biofilms with IBS needs more study. The presence or absence of biofilms, the extent of any biofilm present, and the location in the colon need correlation with the category of IBS and the severity of symptoms [106]. In addition, the predominant bacterial species in stool specimens and the presence and concentration of biochemicals, such as lactic acid and bile salts, should be determined.

## 8. Clostridium Difficile Colitis

*Clostridium difficile* is a spore-forming, Gram-positive anaerobic bacillus which causes *C*. *difficile* infection (CDI) secondary to toxin production. This gastrointestinal disorder is much different from IBD and IBS; its clinical presentation ranges from mild to severe diarrhea, which can be life-threatening. *C*. *difficile* infection is a leading cause of antibiotic-associated diarrhea and healthcare-associated infection globally [142]. The common risk factors for CDI include prolonged antibiotic use and older age. Antibiotics eliminate the intestinal microbiota and allow germination of *C. difficile* spores, which in turn promotes the proliferation and toxin production by *C difficile*. While *C. difficile* spores are important in developing persistent CDI, the formation of biofilm occurs during the interaction between *C. difficile* and colonic biofilm microbiota. This biofilm acts as a reservoir for *C. difficile* and is one of the risk factors for recurrent CDI. The formation of biofilm is a complicated and multifactorial process in vitro, and not every strain of *C. difficile* can form biofilm. Dapa studied the factors that modulate biofilm formation by *C. difficile.* This study was performed in vitro with two *C. difficile* strains (630 and R20291) isolated from an outbreak [143]. This study hypothesized that these biofilms may act differently in the intestinal gut and can persist in the gut. One mechanism includes a mutation in flagellin, as flagella contribute to the architecture of mature biofilms. Other mechanisms include the production of major modulators of quorum sensing-autoinducer 2 (AI-2), the S-layer, germination receptor Slec, and sporulation, which are involved in matrix gene expression and regulate biofilm formation [143].

James et al. studied the ability of different antibiotics to penetrate *C. difficile* biofilms grown in vitro using scanning electron microscopy and showed that surotomycin and fidaxomicin are effective in disrupting *C. difficile* biofilms, but vancomycin and metronidazole had no observable effect [144]. Another study demonstrated the resistance of vancomycin and fecal microbiota transplantation in patients with biofilms [145].

The interaction of *C. difficile* and intestinal biofilms is a driving factor underlying chronic infection and could function as a protective layer for the bacteria against antibiotic exposure. In vitro, *C. difficile* forms aggregates in an extracellular matrix and can interact with other bacterial species in the intestine to increase biofilm formation. Biofilm-associated *C. difficile* cells undergo metabolic remodeling and, compared with planktonic cells, have different cell-surface proteins and organelles. In vivo, *C. difficile* biofilm structures have been observed adjacent to epithelial cells and necrotic microvilli of CDI [145]. The interaction between *C. difficile* and commensal bacterial in the gut has been studied. For example, *C. scindens* and *Fusobacterium nucleatum* can induce and increase *C. difficile* biofilm formation [146,147].

## 9. Management of Colonic Biofilms

Surface-associated bacteria and possibly biofilm-associated bacteria in a normal healthy colon have important functional activities. Changes in the bacterial population in biofilms and the structure and integrity of the biofilm can contribute to the development of inflammatory diseases, such as IBD, and of infections, such as *C. difficile* colitis [3]. Alternatively, these disorders can cause changes in the biofilm, and the altered biofilm merely provides an indication of ongoing inflammation or infection. Clinical efforts to improve the quality of the biofilm or to eliminate abnormal biofilms have potential benefits in patients with both acute and chronic colonic diseases. However, the difficulties in determining the structure and composition of biofilms in the human colon make these projects quite difficult.

Eradication of pathogenic biofilms has proven a challenging task. Typically, most strategies involve clinically approved antibiotics, but other novel compounds have also been tested (see Section 9.3). Results vary, with some antibiotics being wholly ineffective, others causing reductions in the biomass [148,149], and some even inducing biofilm formation [150,151]. Eradication in patients is challenging and, in some cases, may be impossible with current therapies due to drug toxicity and poor biofilm penetration. In vitro experiments, while showing promise for eradication [152], are limited by their ability to replicate the complexity of the gastrointestinal tract. Treatment for colonic biofilms is even less understood, in part because colonic environments include a microbiota with interactions among various species, variations in pH and nutrient levels, interactions with the host immune system and inflammatory responses, the presence of mucus, toxins, and enzymes, and anatomical features, including crypts and diverticula, which may complicate treatment approaches.

Studying the effectiveness of anti-biofilm therapies in human colons is doubly challenging as the biofilm burden will need to be assessed by colonoscopy. Molecular markers for biofilm formation or eradication would be useful; however, to our knowledge, these have not been effectively or routinely used in clinical studies or practice. Lastly, the prevention of future biofilm formation by pathogenic bacteria must also be examined. It is well established that microbiota disruption by antibiotics can lead to disease states, such as *C. difficile* infections. Regulation of healthy microbiota function with probiotics has been studied, but the human gut microbiome is still poorly understood. However, prevention of new pathogenic biofilm formation post eradication is likely to be complicated due both to the complexity of the microbiome itself and the variety of proposed catalysts for pathogenic biofilm formation. Future studies will need to incorporate the complexity of the gastrointestinal tract into the study either by complex in vitro systems, murine models, or human trials.

### 9.1. Antibiotics

A small number of studies have specifically examined antibiotic treatment for colonic biofilms. An in vitro model used a clever arrangement of interconnected chemostat vessels with differing pH and nutrient composition to simulate the human colon. This model demonstrated that not only did *C. difficile* spores associate with existing biofilms but they also could not be eradicated with vancomycin or fecal microbiota transplants [145]. This may be, in part, due to the type of treatment used. For example, fidaxomicin has become the preferred treatment for *C. difficile* over the traditional use of vancomycin and metronidazole. In vitro studies have shown that fidaxomicin is superior to these drugs in preventing biofilm formation [144,153]. Indeed, the persistence of *C. difficile* within biofilms may be responsible for recurrent *C. difficile* [145,154]. Furthermore, reducing but not eliminating biofilms risks disease relapse; one study noted that mucosal bacterial populations suppressed with antibiotic treatment can rebound after cessation of treatment [155]. Maintaining suppression with lifelong antibiotic therapy would be undesirable except in situations of severe illness. Even if and when successful eradication methods are developed, tests of cures will be needed to confirm eradication.

Antibiotic treatments may paradoxically promote biofilm formation. This is due to the sub-minimal inhibitory concentration (MIC) of antibiotics, which produce sufficient stress to induce genes related to biofilm production without killing the bacteria. An alternative theory suggests that extracellular DNA released by bacteria in the population most susceptible to the antibiotic induces the remaining bacteria to form biofilms [156]. This phenomenon has been observed across bacterial species and with various antibiotic classes. The class of antibiotics may be important here, since different antibiotics can induce different gene expression patterns in bacteria related to biofilm formation [157]. The dose range and concentration for which biofilms are stimulated also vary across antibiotics [158]. Of note, sub-MIC concentrations do not always promote biofilm formation. For example, azithromycin inhibits *P. aeruginosa* biofilm formation at sub-MIC levels [158]. Other antibiotics may have nonlinear (multiphasic) dose–response curves with regard to biofilm stimulation and inhibition [158]. In vitro studies suggest that early treatment strategies may help prevent biofilm formation, and this strategy has been used to prevent pulmonary biofilm formation in patients with cystic fibrosis [159]. The age of the biofilm and timing of treatment may also be important. One study noted that treatment of early or “immature” biofilms resulted in better clearance compared to more established biofilms [160].

### 9.2. Probiotics

Disruption of colonic biofilms has been shown to produce disease states. Therefore, strategies to alter or “restore” a healthy microbiome composition have been studied. One method is through the use of supplemental commensal bacteria or “probiotics”. The introduction of probiotics may prevent, for example, diarrheal illness (including *C. difficile* infection) in patients on antibiotic therapy [161,162]. Probiotic bacteria produce these effects via multiple mechanisms, including the production of antimicrobial substances, prevention of pathogenic bacterial colonization or overgrowth by occupying surfaces, alteration of environmental pH, the consumption available nutrients, and toxin elimination [19,163,164]. For example, exopolysaccharides released from *Lactobacillus acidophilus* A4 inhibited enterohemorrhagic *Escherichia coli* biofilms by up to 94% in vitro [165]. A cell-free supernatant from *B. subtilis* KATMIRA19 inhibited *Salmonella* biofilm formation up to 56.9%. Interestingly, there was no effect on planktonic cells, and the authors proposed that these cell-free supernatants may have inhibited *Salmonella* quorum sensing. Another study focused on the compound reuterin, produced by *L. reuteri* using glycerol, in the prevention of *S. mutans* biofilms; although this study focused on oral health, the authors noted that the natural presence of glycerol in the gut creates conditions for reuterin production [166]. Fungal biofilms may also be prevented by probiotics. For example, *Lactobacillus* strains were able to reduce *C. albicans* biofilm formation by up to 61.8% in in vitro studies [167].

Genetically modified probiotics may also be an important tool in eliminating or restoring biofilms. One study engineered *Escherichia coli* Nissle 1917 to produce an anti-biofilm enzyme dispersin B (DspB), as well as anti-pseudomonal bacteriocins S5 pyocin and E7 lysis proteins. DspB works by destabilizing biofilms by hydrolyzing 1,6-N-acetyl-D-glucosamine. The administration of this antibiotic in murine models for gut infection was effective at clearance and prophylaxis of *P. aeruginosa*, and the combination of S5 and DspB was better than S5 alone [168]. Delivery methods of probiotics may also be important. Ironically, biofilm-coated probiotics could help with the delivery of these organisms to the colon [169].

Despite their promise, studies on probiotics and biofilms are limited by a lack of in vivo studies. Exceptions to this include a *C. elegans* and murine model for gut infection, [168]. However, most probiotic studies rely on in vitro testing, and clinical studies are limited. Furthermore, most studies on probiotics and biofilm report success in biofilm prevention but not eradication. A rare exception to this involves *Lactobacillus rhamnosus* ATCC 7469 and *L. plantarum* 2/37, which were able to replace pathogenic *Staphylococcus* biofilms with their own biofilms [170]. This can address the need for biofilm prevention in patients who are decolonized or are at a higher risk for biofilm-associated GI illnesses, but it does not address the current challenges of eradication. Lastly, bacteria from a variety of sources have been examined for antibiofilm properties, including human fecal matter [171] and fermented foods [172,173]. This highlights the immense microbiological diversity and the possibility for the discovery and research of new probiotic strains [174].

### 9.3. Other Therapies

A variety of additional therapies have been proposed, including nanoparticles, matrix-degrading enzymes, quorum sensing inhibitors, and antimicrobial lipids [175,176]. A review of these compounds can be found in articles by Sharma, Misba, and Khan [175], and Verderosa, Totsika, and Fairfull-Smith [176].

## 10. Critique

The colon has a very large surface area and volume, and it contains a large number of microorganisms, including bacteria, viruses, archaea, and fungi. These microorganisms may be attached to the surface of the colon or in luminal contents. Some bacteria form biofilms, but the frequency of biofilm formation in the healthy normal colon is uncertain. In addition, the types and number of bacteria attached to the normal colon surface are uncertain. Most patients undergoing colonoscopy have undergone bowel preparation procedures, which potentially reduce the number of bacteria attached to the colon. Therefore, determining whether normal biofilms are present requires a noninvasive method to identify these biofilms. This might involve the detection of quorum sensing molecules and/or the detection of bacteria that are usually present in biofilms. To date, these methods are not routinely available. Consequently, all studies involving healthy control subjects and patients with colonic diseases potentially provide information that is not truly representative of the actual in vivo situation.

Endoscopists have an excellent opportunity to contribute to studies on biofilm formation and their association with gastrointestinal disorders. Simple inspection of the colon can identify biofilms and map their location in the colon [105]. Small mucosal biopsies would provide specimens for histologic studies, scanning electron microscopy studies, and bacterial identification. This would require coordination with specialized centers which have the technological expertise to conduct these assays.

## 11. Concluding Remarks

The human colon has a complex anatomy and physiology. The microbiome includes hundreds of bacterial species and a large cellular mass. These bacteria can be free-floating in the stool, attached to particulates in the stool, attached to the mucous layer of the epithelial surface, or organized into biofilms on the mucosal surfaces. The commensal bacteria in the colon provide important nutritional support for the colonic epithelium by metabolizing undigested fibers and polysaccharides to produce short-chain fatty acids. Butyric acid is then used in mitochondria to produce ATP. Multiple factors can alter biofilms in the colon; these factors include antibiotics, bacterial infections with *C. difficile*, and acute and chronic intestinal inflammation. Altered or abnormal biofilms have been associated with both malignant and benign colonic diseases. The important question is whether an abnormal biofilm stimulates the development of a benign disease, such as inflammatory bowel disease, or whether the inflammatory bowel disease creates the abnormal biofilm that may or not contribute to the pathogenesis. Another important question is whether or not restoring or creating a normal biofilm improves colon health and prevents the development of either malignant or benign disease. These questions cannot be easily answered until noninvasive methods become available to monitor biofilm mass and composition.

## Figures and Tables

**Figure 1 ijms-23-14259-f001:**
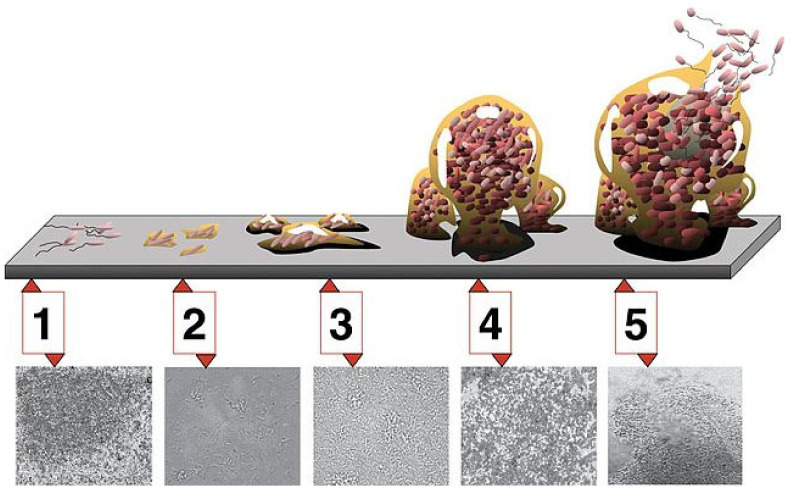
Stages in the development of a biofilm. Stage 1 involves adherence, stage 2 involves attachment and replication, stage 3 involves microcolony formation, stage 4 involves the production of the matrix with biofilm formation, and stage 5 involves partial dispersion of the biofilm, with the free-floating bacteria potentially forming new biofilms [5].

**Figure 4 ijms-23-14259-f004:**
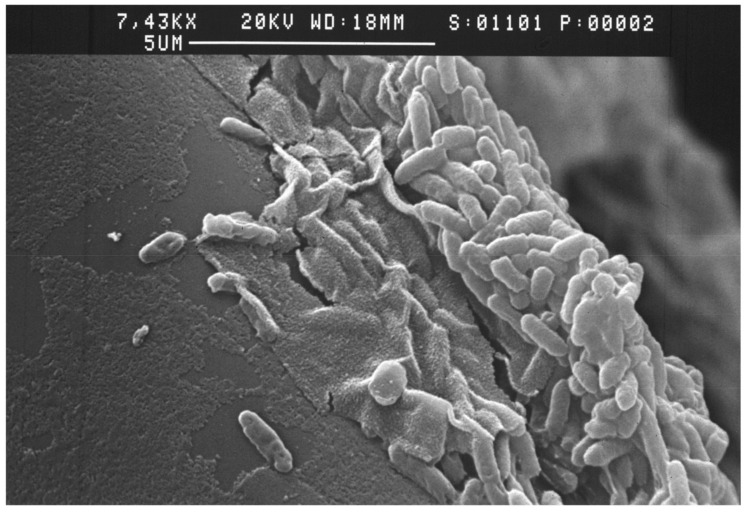
Formation of *P. aeruginosa* biofilms on glass wool as observed by scanning electron microscopy. This image demonstrates cellular aggregates and the matrix [71,72].

## Data Availability

Not applicable.

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
