# Peer review of "Biofilms and Benign Colonic Diseases"

_ijms, 2022, doi:10.3390/ijms232214259_

Round 1
Reviewer 1 Report
This review of biofilms and their association with colonic disease states is an important and intensively studied area of research. A vast literature exists and several recent review articles have been published on different aspects of this field. This manuscript errors, in my opinion, by being overly broad in its scope and thus lacking depth on any singular subtopic. As one example of this, short sections are presented on various methodologies to measure biofilms (section 4.1), which is not a complete listing and provides very little actual detail on each of the methods. Why is this part of this particular review, especially when the point is repeatedly made that in vitro approaches are of little value. The section 2 biofilm formation focuses on Pseudomonas aeruginosa, a species that is of little consequence in diseases of the colon. Perhaps a more relevant pathogen would be more appropriate for this topic. The review is lengthy, more than 800 lines of text not including the reference section). The most important sections for the stated topic of the review are sections 5-9. Although this is the most interesting component of this review, I found it difficult to get through. The authors provide a series of examples of studies and state the findings as a list of bacterial species comprising the biofilms and/or percentages of various subcategories in healthy individuals versus patients with different clinical conditions. This becomes a “wall of words and numbers” and the take away message becomes lost. The major limitation of the review is that there is no overarching analysis of the existing literature to provide the readers with a conceptual framework of the state of the field other than it is complex, poorly understood, and must be further studied in vivo. Sections 10 and 11 should be where this critical analysis would be found, but these were very short and not altogether helpful.
Minor points:
Formatting errors (which may have occurred when the text was converted to pdf format) was a problem, especially with the References. Numerous grammatical errors were present as were undefined abbreviations.
Specific comments:
1. Line 9: the 12 should be an exponent value.
2. Lines 105-108: Pel and Psl are undefined abbreviations.
3. Lines 108-113: It is not clear how antibiotic penetration into a biofilm models nutrient penetration. Are there examples of nutrient distribution similar to the two examples of antibiotic distribution given here? Does tobramycin have high affinity to cells which would explain the peripheral distribution pattern?
4. Line 119: GMP is the standard abbreviation of the guanine nucleotide, not the di-nucleotide. Also, “cells” should be deleted from this sentence.
5. Lines 119-120, need a reference citation for this sentence.
6. Lines 120-126: Reference 9 is a micro-review that briefly mentions the T4 pili of Pseudomonas but does not mention the WSP system or the CyaB protein. It would be better to cite specific Pseudomonas references for these two adhesion mechanisms.
7. Lines 130-132: Reference citations needed here.
8. Lines 138-144: Reference citations needed here.
9. Lines 162-165 and 170: Reference citations needed here.
10. Figures 1, 2, and 4 legends: These are not helpful. They provide the original citation, but no other detail. Therefore, it is not clear why these are specifically included in the review.
11. Lines 174-176: The summary is accurate for the adhesion aspect, but no information is provided for the other aspects of biofilm formation.
12. Line 184: why no reference for sporulation as was done for the other phenotypes? Is sporulation a relevant aspect of Pseudomonas-involved mixed biofilms?
13. Lines 225-229: It is unclear what 3-oxo-C21 is or what the listed bacteria have to do with Pseudomonas biofilm formation.
14. Lines 292-294: Reference citations needed here.
15. Lines 303-304: Should be “electron” microscopy.
16. Lines 306-307: CLSM is not technically electron microscopy.
17. Figure 4: This is an SEM of a mixed culture biofilm. Is Pseudomonas even involved in this biofilm. If not, why not show an SEM of an actual Pseudomonas biofilm?
18. Line 334: Since CRA is the abbreviation of Congo red agar, following it with agar is redundant.
19. Lines 336-337: since “such as” implies a nonexclusive set of examples, “etc.” is not needed here.
20. Line 340-347: or stained with crystal violet. Should also mention that the amount of biofilm can be quantified through solvent extraction of the dye and spectrophotometric analysis (as indicated in the following microplate assay).
21. Lines 413-430: Reference citations needed here.
22. Lines 432-438: What were the results with the samples from patients with UC or self-limiting colitis and what conclusion can be drawn from this?
23. Lines 462-463: How determined (from genomic analysis or metabolic assays)?
24. Lines 474-487: Reference citations needed here.
25. Lines 495-501: Reference citations needed here.
26. Lines 526-528: CD or IBD patients?
27. Lines 529-530: Provide citations for these studies, or are you referring to references 83 and 101?
28. Line 562: Delete “identified mucosa”.
29. Line 566: Not sure what you mean by “similar mechanism” in this context.
30. Lines 578-580: A better explanation should be provided as to how an extracellular biofilm controls intracellular communities.
31. Lines 701-703: What type of novel compounds (no examples or citations are provided).
Author Response
Please see attachment with responses to each comment

Reviewer 2 Report
The review ''Biofilm and benign colonic diseases'' is well written, balanced and explicit. Small changes in a deffinition and sugestion to modify the References accourding to the template.
I suport publication after minor changes.

Author Response
We have made changes in the glossary as suggested in your review of our article. Thank you for your help.
Round 2
Reviewer 1 Report
The authors corrected most of the citation deficiencies, figure legend issues, and problems with the abbreviations. The manuscript is improved because of these edits. The major issues, the lack of focus and resulting lack of depth of the review, and the lack of perspective and analysis of the literature to date remain. It remains unclear in my mind why there is such an emphasis on Pseudomonas aeruginosa in this review, when it is not a major component of colonic biofilms. There is a great deal of information known regarding biofilm formation by this bacterium, but the same can be said of more relevant gut bacteria.
New specific comments:
1. Lines 309-312: In the paragraphs above this, you indicate that QS AHLs are present in stools of healthy individuals as patients with IBD or in remission. So how do you reach the conclusion that “The detection of QS in stool specimens would provide evidence of the presence of biofilms in the colon, and the concentration of QS should indicate the amount of biofilm present.”? Actual evidence, or citations of reports making this claim need to be provided. After all, QS systems are active in a great deal of regulatory activities unrelated to biofilm formation.
Author Response
November 5, 2022
Dear editor,
We have reviewed the comments by your reviewer carefully and have made changes in the manuscript to reflect these comments. This reviewer states that there is a lack of focus and a resulting lack of depth of the review. Therefore, there is a lack of prospective and analysis of the literature to date. In our view, we have reviewed biofilm formation in some detail. Critical features of the formation of a biofilm which greatly depends on the ability of the bacterium to attach to surfaces and quorum sensing. Those topics were reviewed in detail. We did use Pseudomonas as a key bacterium in that part of the review. We have added a section to the paper that now includes information about Bacteroides fragilis. This is an important gut bacterium which is potentially relevant to both the health of the colonic epithelium and is potentially relevant to disease states associated with Bacteroides infections. We had a section on factors which may influence the composition and stability of biofilms. We added additional prospective and analysis. This is a relatively complicated topic that is hampered to a great extent by the inability to sample biofilms in healthy subjects and patients. In addition, multiple factors influence of formation and stability of biofilms and many studies focus on only 1 or 2 of these factors. Very few studies look at serial changes in biofilm which seem essential to understand their role in normal colonic health and disease.
The speculation regarding the presence of QS AHKLs in healthy individuals and in patients reflects the idea that some healthy individuals have biofilms but the total mass of biofilms is probably lower than in some patients with colonic diseases. It is possible that the measurement of QS in stool specimens would provide useful information if the levels vary between healthy subjects and patients and if the levels in patients vary over time according to the disease course. We will modify that statement.
This second revision includes approximately 1170 new words and 11 new references. We are submitting a revised document using track changes. However, I accepted all the changes in the first revision to create a new clean document. The track changes in the current second revision reflect only the changes in the second revision.
If we need to make additional changes, please contact us.
Respectfully yours,
Kenneth Nugent
Round 3
Reviewer 1 Report
The manuscript is improved over the previous versions.